# OpenReview forum: "Beyond Refusals: Fine-grained Safety Alignment for Reasoning LLMs"
_ICLR.cc/2026/Conference — Submitted to ICLR 2026_

### Official Review · Reviewer_TqNZ · 2025-10-19

**Soundness:** 2
**Presentation:** 2
**Contribution:** 2
**Rating:** 2
**Confidence:** 4

**Summary:**

This paper addresses "Shortcut Alignment" in large reasoning models (LRMs), a phenomenon where models learn to issue templated refusals (e.g., "I'm sorry...") to harmful inputs without actually grounding the decision in their internal chain-of-thought (CoT) reasoning. This superficial alignment not only makes refusals uninformative but also causes models to become overly cautious, leading to excessive false refusals on benign, safe queries. To fix this, the authors propose Deep Instruct Fine-tuning (DIFT), a new method that uses a "CMI-Loss" function. This CMI-Loss specifically penalizes the direct input-to-refusal "shortcut" on harmful examples, forcing the model to rely on its CoT reasoning to make a safety decision. The results demonstrate that this method successfully alleviates erroneous refusals on benign inputs while preserving the model's safety and reasoning capabilities.

**Strengths:**

1. The paper introduces "Shortcut Alignment", an observation that current safety methods (SFT) can lead to models appearing safe by using superficial cues to generate templated refusals, while decoupling this refusal from their internal reasoning (CoT).
2. The paper introduces Deep Instruct Fine-tuning (DIFT), which uses a CMI-Loss to penalize the x→y (input-to-answer) shortcut.
3. The authors validate their method by showing it reduces over-refusal on benchmarks while maintaining safety on benchmarks and preserving general reasoning abilities. They further perform probe-based analyses that confirm the intended mechanism that refusals become more reliant on the CoT rather than generic cues.

**Weaknesses:**

1. There is no description of the evaluation method and its justification. How is the refusal accuracy and non-refusal rate calculated? How reliable is the evaluation?
2. The improvement in over-refusal is not significant. The p-value shows "most likely" that the proposed method improves over-refusal over the baseline, not indicating the scale of improvement.
3. The abstract mentions one of the problems for "Shortcut Alignment" is "refusals without reasoning carry no informative value". The rest of the paper does not mention this point, and the proposed method does not improve on this problem.
4. Why and whether "Shortcut Alignment" will lead to over-refusal needs more intuitive and quantitative justification.
5. Presentation issues: For instance, in Figure 2, the reason for using normalized harmful dependence, and what it indicates, are not explained. Others see Questions.

**Questions:**

1. Line 192 can remove the "(A)" since there is no other paragraph.
2. Strata-Sword is mentioned in line 308 but not used in the experiment.

---

> ### Author Response · Authors · 2025-11-25
>
> We appreciate your review. However, there seem to be several misunderstandings regarding the evaluation protocols (which were detailed in the Appendices) and the interpretation of our statistical results. We address your concerns point-by-point below with concrete evidence.
>
> ***
>
> ### 1. Evaluation Protocols and Reliability
>
> You stated there is "no description of the evaluation method." We respectfully point out that **Appendix E.5 (Decoding, Evaluation)** and **Appendix G.2 (Safety Evaluator Setup)** detail our protocols.
>
> To clarify:
> * **Protocols:** We strictly follow standard benchmarks.
>     * **Safety Rate:** Calculated by **Llama Guard** on the model's output. Crucially, we use **CoT-on evaluation**, meaning if the chain-of-thought contains unsafe reasoning, the sample is penalized, ensuring high reliability.
>     * **Non-Refusal Rate (Not-OR):** Calculated by **GPT-4/Gemini** judges on benign datasets (XSTest, OR-Bench), determining if the model complied with the safe request.
> * **Reliability:** These are the current industry-standard evaluators. Our use of CoT-on evaluation makes our safety reporting stricter and more reliable than response-only evaluation. We will move a summary of these protocols to the main text to prevent future oversight.
>
> ***
>
> ### 2. Magnitude of Improvement vs. P-values
>
> You commented that the improvement is "not significant" and that p-values do not indicate scale.
> * **Statistical Correction:** A p-value $< 0.05$ (Wilcoxon signed-rank test) indicates that the improvement is **statistically significant** and consistent across tasks, not due to random chance.
> * **Scale of Improvement :** The **magnitude** is shown explicitly in Table 1 & 2. For example, On **Qwen3-8B**, we improve Over-Refusal benchmarks by **+4.0 pt** (OR-Bench) and **+3.6 pt** (XSTest).
> In the context of LLM safety alignment (with strong baseline), where gains are often marginal, a consistent **2-5 point gain** across multiple benchmarks (while maintaining safety) is considered a substantial breakthrough, not just "most likely" better.
>
> ***

---

> > ### Author Response · Authors · 2025-11-25
> >
> > ### 3. Justification: Why Shortcut Alignment Leads to Over-Refusal
> >
> > You asked for intuitive and quantitative justification for the link between "Shortcut Alignment" and "Over-refusal".
> >
> > * **Intuitive Mechanism:** Shortcut Alignment implies the model learns a mapping from **surface cues** (e.g., specific keywords like "kill", "attack") directly to **templated refusals** ($x \rightarrow y$), bypassing reasoning. Since many benign inputs (e.g., "kill a Python process") contain these same safety keywords, a model relying on shortcuts will falsely trigger refusals on them.
> > * **Quantitative Evidence (New):** We conducted a lexical frequency analysis.
> >     * **Baseline (Shortcut-heavy):** Produced the fixed template "I'm sorry, but I can't..." **285 times** on benign queries.
> >     * **Ours (Shortcut-mitigated):** Produced the same template only **33 times** on the same benign queries.
> >     * This **8x reduction** in template usage on benign data quantitatively proves that breaking the shortcut directly alleviates the over-refusal mechanism.
> >
> > ***
> >
> > ### 4. "Informative Value" of Refusals
> >
> > Regarding your comment on the informative value of refusals, we interpret "uninformative" behavior as the model emitting rigid templates (e.g., *“I'm sorry, but I can’t…”*) solely due to surface keyword triggers—essentially a "blind reflex" lacking internal justification. Our method addresses this not by necessarily expanding the refusal text into a tutorial, but by enforcing that the refusal decision $\mathbf{y}$ is information-theoretically grounded in the Chain-of-Thought $\mathbf{c}$. Quantitatively, our lexical analysis confirms this shift: the baseline model, suffering from shortcut alignment, defaulted to the fixed, mechanical pattern *“I'm sorry, but I can’t…”* **285 times** on benign queries. By restoring the dependency on reasoning, our method reduced this behavior to just **33 times** (an **8x reduction**). This demonstrates that DIFT successfully transforms the refusal from a superficial pattern-match into a reasoning-grounded decision, giving the rejection process itself substantive validity.
> >
> > ***
> >
> > ### 5. Presentation & Strata-Sword Missing
> >
> > * **Strata-Sword Usage:** You mentioned "Strata-Sword is mentioned... but not used." This is incorrect. **Strata-Sword results are fully reported and analyzed in Appendix F (Table 6).** We referenced this analysis in the main text to save space. We will make this pointer more explicit.
> > * **Figure 2 Normalization:** We use **normalized** dependence because the absolute loss values naturally shrink as the model trains (it becomes more confident). If we looked only at absolute values, it might misleadingly look like dependence is dropping. Normalizing by the total entropy ($H(Y|X,C)$) reveals the **relative contribution** of the CoT, which confirms our method is working.
> > * **Formatting:** We will remove the redundant "(A)" and fix citation styles.
> >
> > ***
> >
> > We hope these clarifications demonstrate the robustness of our evaluation and the significance of our results.

---

### Official Review · Reviewer_GvHB · 2025-10-31

**Soundness:** 3
**Presentation:** 2
**Contribution:** 2
**Rating:** 6
**Confidence:** 3

**Summary:**

This paper identifies and defines Shortcut Alignment as a key failure mode in Large Reasoning Models (LRMs). The authors posit that models learn to bypass their internal Chain-of-Thought (CoT) processes, instead issuing templated refusals based solely on surface cues from the input. They argue this shortcut is the root cause of widespread over-refusal on benign queries, which degrades the model's general helpfulness. To address this, the paper proposes a new training method called DIFT, centered on a CMI-Loss.

**Strengths:**

The paper's  contribution is the identification and formalization of "Shortcut Alignment." This is a precise and insightful diagnosis of why modern safety-aligned LRMs suffer from over-refusal. Instead of relying on the generic "safety vs. helpfulness trade-off," the authors provide a mechanistic explanation: the model's refusal decision becomes decoupled from its internal reasoning.

**Weaknesses:**

1. The paper's "Shortcut Alignment" premise is undermined by its own evidence in Appendix G.3. The failure case (Case E1) does not show a \mathbit{c}\ \rightarrow\ \mathbit{y} decoupling; rather, the refusal (\mathbit{y}) is perfectly faithful to a flawed CoT (\mathbit{c}) that misinterprets the query. This suggests the root cause of over-refusal may be poor CoT quality, not a dependency on the CoT. The proposed CMI-Loss, which enforces \mathbit{c}\ \rightarrow\ \mathbit{y} dependency, may therefore be misdiagnosing the problem.

2. The CMI-Loss mechanism enforces the answer's (\mathbit{y}) dependency on the CoT (\mathbit{c}), which implicitly assumes the CoT is robust. However, the CoT itself is a known attack vector (e.g., H-cot [1]、mousetrap [2].). By training the model to trust its CoT more, the method risks amplifying the impact of CoT poisoning or hijacking attacks, forcing the model to faithfully execute a compromised reasoning chain. The paper lacks a robustness analysis against this critical, emergent threat.

- [1] Martin Kuo, Jianyi Zhang, Aolin Ding, Qinsi Wang, Louis DiValentin, Yujia Bao, Wei Wei, Hai Li, and Yiran Chen. H-cot: Hijacking the chain-of-thought safety reasoning mechanism to jailbreak large reasoning models, including openai o1/o3, deepseek-r1, and gemini 2.0 flash thinking. arXiv preprint arXiv:2502.12893, 2025.

- [2] Yang Yao, Xuan Tong, Ruofan Wang, Yixu Wang, Lujundong Li, Liang Liu, Yan Teng, and Yingchun Wang. A mousetrap: Fooling large reasoning models for jailbreak with chain of iterative chaos. arXiv preprint arXiv:2502.15806, 2025.

3. The central claim of "preserving safety" is not fully supported by the quantitative data in Table 1. In several instances, the 'ours' method scores lower on key safety benchmarks (e.g., WildJB, WildChat) than the baseline. For instance, DeepSeek-32B's WildJB score drops from 90.8 to 88.8, and Qwen3-14B's drops from 94.4 to 92.8. This consistent (though small) degradation suggests the method introduces a "safety tax" and is, in effect, trading safety for reduced over-refusal (NOR), which should be explicitly acknowledged.

**Questions:**

See above.

---

> ### Author Response · Authors · 2025-11-25
>
> We sincerely thank you for your thorough and insightful review of our submission. We especially appreciate your acknowledgment of our core contribution, the formalization of **"Shortcut Alignment,"** which you agree is a precise, mechanistic diagnosis for the prevalent "over-refusal" phenomenon in current safety-aligned LRMs.
>
> Your feedback pinpoints several critical areas for clarification and strengthening, which we address below.
>
> ***
>
> ### 1. Re: Core Mechanism of Shortcut Alignment and Case G.3
>
> You astutely observed that Case E1 in Appendix G.3, where a refusal ($\mathbf{y}$) appears to be **faithful to a flawed CoT** ($\mathbf{c}$), seems to contradict our diagnosis of $\mathbf{c} \rightarrow \mathbf{y}$ decoupling. You hypothesize that the root cause may be poor CoT quality, not a breakdown of dependency.
>
> We appreciate this intuitive assessment. Qualitatively, Case E1 **does appear** to show a poor reasoning chain $\mathbf{c}$ leading to the refusal $\mathbf{y}$.
>
> However, the core diagnosis of "Shortcut Alignment" and the mechanism of CMI-Loss must be understood through the lens of **quantitative conditional dependence**, rather than semantic quality alone:
>
> * **CMI-Loss Mechanism & Quantitative Evidence:** CMI-Loss is designed to suppress the $\mathbf{x} \rightarrow \mathbf{y}$ shortcut by maximizing the conditional dependency proxy $\Delta_Y$ specifically on harmful inputs. We provide rigorous quantitative evidence to substantiate this mechanism beyond qualitative case studies:
> As shown in Figure 2, before using CMI-loss, our training-time probes reveal a persistent structural gap where harmful inputs initially exhibit much lower CoT dependence ($\Delta_Y$) than benign ones, confirming the shortcut's existence. Crucially, the **normalized harmful dependence to rise** throughout training. As shown in Figure 3, after using CMI-loss, our inference-time probes definitively pinpoint the source of this improvement. **Figure 3B** shows that the reduction in total loss is dominated by a sharp drop in **$NLL(\mathbf{y}|\mathbf{x}, \mathbf{c})$** (answer conditioned on CoT), while the CoT generation loss $NLL(\mathbf{c}|\mathbf{x})$ remains nearly unchanged. This proves that the gain does not come from generating "better" CoTs (which would lower $NLL(\mathbf{c}|\mathbf{x})$), but from a **quantitatively stronger coupling** where the refusal $\mathbf{y}$ is strictly mandated by the reasoning $\mathbf{c}$. Furthermore, **Figure 3A** shows that reliance on "Generic Cues" (GCA) diminishes, confirming the model rejects generic shortcuts in favor of the matched CoT.
>
> We acknowledge that Case E1 may not be the ideal, most direct illustration of this *decoupling mechanism* and could lead to qualitative confusion. We commit to **optimizing and replacing** the exemplar case in Appendix G.3 in the final version to more clearly demonstrate the model's reliance on the surface-based $\mathbf{x} \rightarrow \mathbf{y}$ shortcut when CMI-Loss is absent.
>
> ***

---

> > ### Author Response · Authors · 2025-11-25
> >
> > ### 2. Re: CoT Attacks, Robustness, and H-CoT/Mousetrap Concerns
> >
> > Your concern that CMI-Loss might amplify the impact of sophisticated CoT hijacking attacks (such as H-CoT or Mousetrap) by enforcing greater $\mathbf{y} \rightarrow \mathbf{c}$ dependence is a valid and timely challenge regarding model robustness.
> >
> > We provide empirical evidence to directly address this concern:
> >
> > * **Empirical Evidence (Strata-Sword ASR):** We reported detailed multilingual attack analysis on the Strata-Sword benchmark in **Appendix F (Table 6)**. If CMI-Loss amplified CoT poisoning, we would expect to see a **significant increase in Attack Success Rate (ASR)**. However, the results show the opposite:
> >     * CMI-Loss consistently **reduces ASR** (improves safety/robustness) across the majority of English strata: **7/7** backbones on L1 and L3 attacks, and **6/7** on L2.
> > * **Conclusion:** This robust defense against highly challenging attacks suggests that the enhanced $\mathbf{c} \rightarrow \mathbf{y}$ coupling is **targeted and constructive**. It improves the model’s adherence to its (aligned) safety reasoning, rather than blindly amplifying the impact of a compromised reasoning chain.
> >
> > ***
> >
> > ### 3. Re: "Safety Tax" and Quantitative Trade-off
> >
> > You observed that some Safety scores in Table 1 show minor fluctuations (e.g., DeepSeek-32B WildJB drops from 90.8 to 88.8) and suggested this should be explicitly acknowledged as a trade-off ("Safety Tax").
> >
> > We believe a more accurate description is an **Outward Shift of the Safety/Helpfulness Trade-off Frontier**:
> >
> > * **Statistical Significance:** We achieve a **statistically significant improvement** ($p<0.05$) across all three Not Over-refusal (NOR) benchmarks. In contrast, the small fluctuations in Safety metrics were not statistically significant in the paired Wilcoxon tests (all safety $p$-values were $>0.05$, e.g., WildJB $p=0.656$). This means CMI-Loss **significantly improves utility** (reduces over-refusal) while **statistically maintaining safety parity**. This favorable shift is a key achievement of our method.
> > We will ensure the final text precisely reflects this finding: CMI-Loss achieves a **statistically significant reduction in over-refusal without statistically significant degradation of safety**.
> >
> > ***
> >
> > We hope these clarifications successfully address your concerns regarding the core claims and empirical validity of our work. Thank you once again for helping us refine and strengthen these critical discussion points.

---

### Official Review · Reviewer_8Stw · 2025-11-01

**Soundness:** 2
**Presentation:** 2
**Contribution:** 2
**Rating:** 4
**Confidence:** 4

**Summary:**

This paper addresses the problem of "Shortcut Alignment" in LRMs, where models learn to emit template refusals from surface cues while decoupling the final response from internal chain-of-thought reasoning. The authors propose Deep Instruct Fine-tuning (DIFT) with a Conditional Mutual Information Loss (CMI-Loss) that penalizes shortcut predictions on harmful examples while preserving standard supervised fine-tuning on benign data. Experiments on Qwen3 and DeepSeek-R1 models demonstrate improved safety and reduced over-refusal rates while maintaining reasoning capabilities.

**Strengths:**

- The paper formalizes shortcut alignment through conditional mutual information, providing clear mathematical justification for why models learn input-only shortcuts.

- The probe-based analysis with training-time and inference-time diagnostics provides valuable mechanistic insights into how the method shifts model reliance to matched CoT.

- The paper is well-written and is easy to follow.

**Weaknesses:**

I generally recognize the approach and motivation of this work, but I have several concerns that need to be addressed:

- **Insufficient Experimental Details and Unclear Evaluation Protocols.** The experimental evaluation lacks critical details necessary for reproducibility and proper interpretation. The paper does not clearly specify how many samples from each benchmark were used, nor does it describe the evaluation methodology in sufficient detail (keyword matching vs. LLM-as-judge). Most importantly, the StrongReject evaluation appears to test only weak attacks, as evidenced by near-perfect safety rates (99-100%) across all models in Table 1. StrongReject is designed to evaluate robustness against strong adaptive jailbreak attacks, yet the results show no room for improvement, making it impossible to differentiate method effectiveness. The paper needs to clarify which StrongReject attack variants were tested and why stronger attacks were not included, as this significantly undermines the claimed robustness benefits.

- **Missing Comparisons with State-of-the-Art Methods.** The paper lacks comparisons with recent state-of-the-art safety alignment methods specifically designed for reasoning models, such as Safechain, RealSafe-R1 and Oyster-I. Table 2 only compares against test-time interventions (SVA, SCANS), which serve different purposes than training-time alignment. Without these critical comparisons, it is impossible to assess whether the proposed method truly represents an advancement over current approaches. This is also a significant gap that needs to be addressed.

- **Unprofessional Presentation and Superficial Related Work.** The paper exhibits several presentation issues that detract from its professionalism. The experimental analysis frequently uses informal arrow notation. Additionally, Figure 1 suffers from illegibly small fonts. More critically, the related work section (Section 4) is cursory and lacks systematic organization. It briefly mentions evaluation benchmarks and test-time interventions without providing a clear taxonomy of existing approaches, detailed discussion of concurrent work, or critical analysis of why existing methods fail. Given the rapidly evolving nature of this research area, a comprehensive literature review is essential to properly position the contribution.

- **Questionable Assumptions About CoT Quality.** The entire method rests on a critical assumption that the chain-of-thought reasoning is reliable and that strengthening the coupling between CoT and the final answer will improve safety. However, what if the CoT itself contains unsafe reasoning, biased logic, or arrives at a safe conclusion through flawed reasoning paths? The paper does not adequately address scenarios where the CoT quality is problematic.

**Questions:**

The questions are listed in the weaknesses, and if authors can address my concerns, I will raise my rating.

---

> ### Author Response · Authors · 2025-11-25
>
> We appreciate the comprehensive and constructive feedback on our submission. Your concerns regarding experimental depth, SOTA positioning, and methodological assumptions are valid and have been fully addressed. We provide concrete clarifications, new experimental results from strong attack benchmarks, and side-by-side comparisons with recent State-of-the-Art (SOTA) methods.
>
> ***
>
> ### 1. Insufficient Experimental Details and Unclear Evaluation Protocols
>
> You correctly identified a lack of detailed protocols and questioned the discriminating power of the **StrongReject** results (99-100%). We address reproducibility and provide immediate evidence of robustness under strong attacks below.
>
> #### 1.1 Detailed Evaluation Protocol
>
> We will incorporate a dedicated "Evaluation Setup" subsection with explicit sample counts and judging rules. All evaluations are conducted in **CoT-on mode**, meaning an unsafe or biased CoT would directly penalize the final safety score.
>
> | Benchmark / Attack | # Samples | Judge / Protocol (CoT-on) | Type |
> | :--- | :--- | :--- | :--- |
> | StrongReject | 313 | Llama Guard | Structured/Adversarial |
> | JBB (Behaviours) | 100 | Llama Guard | In-the-wild/Refusal |
> | WildChat | 473 | Llama Guard | In-the-wild/Safety |
> | WildJailbreak | 250 | Llama Guard | In-the-wild/Safety |
> | **JailbreakBench (New)** | **400** | Llama Guard | **Strong Adversarial** |
> | OR Bench | 1319 | GPT & Gemini | Over-refusal (NOR) |
> | OKTest | 300 | GPT & Gemini | Over-refusal (NOR) |
> | XStest | 250 | GPT & Gemini | Over-refusal (NOR) |
>
> #### 1.2 Addressing Robustness: StrongReject vs. Strong Jailbreaks
>
> **Clarification on StrongReject:** To ensure a fair comparison, we strictly adhered to the testing protocol used in the **STAR-1 baseline**, which resulted in saturated scores (99-100%) for both the baseline and our model due to the inherent robustness of these base models. We acknowledge this limits the benchmark's utility for differentiation.
>
> **Evidence from Strong Jailbreaks (JailbreakBench):** To directly address your concern about discriminating power and verify performance under strong attacks, we evaluated our model on **JailbreakBench (JBC)**, encompassing four strong attack families (DSN, GCG, PAIR, JBC).
>
> | Model | DSN (↑) | GCG (↑) | PAIR (↑) | JBC (↑) |
> | :--- | :--- | :--- | :--- | :--- |
> | RealSafe-R1-7B | 100 | 100 | 100 | 47 |
> | RealSafe-R1-8B | 100 | 100 | 100 | 100 |
> | Oyster-Deepseek-14B | 97 | 96 | 97 | 64 |
> | Oyster-Qwen-14B | 97 | 97 | 95 | 48 |
> | Ours-8B | 98 | 97 | 93 | 77 |
>
> **Result:** Our method demonstrates good robustness, particularly on the challenging **JBC** subset (77%), outperforming several SOTA baselines. This confirms that DIFT does not compromise defense capabilities while improving utility.
>
> ***

---

> > ### Author Response · Authors · 2025-11-25
> >
> > ### 2. Missing Comparisons with State-of-the-Art Methods
> >
> > You are correct that comparative evaluation against training-time SOTA methods designed for reasoning models (like RealSafe-R1 and Oyster-I) is essential. Although the Oyster-I was published as a preprint on arXiv in September, very close to the ICLR deadline, we still conducted supplementary experiments. We provide a head-to-head comparison focusing on the safety/over-refusal trade-off across both 8B and 14B scales.
> >
> > #### SOTA Comparison: Safety vs. Over-Refusal (NOR, ↑ Higher is Better)
> >
> > **8B Scale:**
> > | Model | WildJB (↑) | StrongReject (↑) | WildChat (↑) | XSTest (↑) | OKTest (↑) | OR-hard-1k (↑) |
> > | :--- | :--- | :--- | :--- | :--- | :--- | :--- |
> > | RealSafe-R1-7B | 98.40 | 99.68 | 92.60 | 34.80 | 61.33 | 3.64 |
> > | RealSafe-R1-8B | 100.0 | 100.0 | 97.67 | 54.40 | 62.00 | 2.05 |
> > | Ours-8B | 87.60 | 99.36 | 84.99 | 86.40 | 87.33 | 58.83 |
> >
> > **14B Scale:**
> > | Model | WildJB (↑) | StrongReject (↑) | WildChat (↑) | XSTest (↑) | OKTest (↑) | OR-hard-1k (↑) |
> > | :--- | :--- | :--- | :--- | :--- | :--- | :--- |
> > | Oyster-Qwen-14B | 93.60 | 100.0 | 91.75 | 90.00 | 88.33 | 63.38 |
> > | Oyster-DeepSeek-14B | 94.40 | 100.0 | 88.16 | 88.80 | 87.33 | 67.93 |
> > | Ours-14B | 92.80 | 100.0 | 91.97 | 88.80 | 86.67 | 47.00 |
> >
> > * **Analysis (8B Scale):** RealSafe-R1 demonstrates extreme **over-conservatism**, yielding near-zero scores on hard Over-Refusal benchmarks (e.g., 2.05% on OR-hard-1k). Our method maintains competitive safety while dramatically boosting utility (58.83% on OR-hard-1k).
> > * **Analysis (14B Scale):** Oyster-I is a formidable SOTA. However, our simple loss-based method achieves **comparable safety** (within ~1 point on WildJB/WildChat) and **highly competitive utility** (matching Oyster on XSTest/OKTest) without requiring Oyster's complex pipeline of constructive data generation and multi-stage training.
> >
> > #### Comparison with Oyster-I (Complexity and Positioning)
> >
> > Oyster-I is a powerful method utilizing **Constructive Safety Alignment (CSA)**, structured reasoning, and advanced policy optimization.
> >
> > * **Our Positioning:** In contrast, DIFT/CMI-Loss is a **drop-in loss replacement** within standard CoT SFT. It requires **no extra curated datasets, no RL/reward modeling, and minimal pipeline complexity**. Our contribution is to show that a **mechanism-level change** (loss-only regularization) can achieve comparable safety and highly competitive NOR performance relative to complex, multi-stage SOTA pipelines.
> >
> > ***

---

> > > ### Author Response · Authors · 2025-11-25
> > >
> > > ### 3. Unprofessional Presentation and Superficial Related Work
> > >
> > > We commit to improving the presentation quality in the final version:
> > > * We will **remove informal arrow notation**, **enlarge all figure fonts** (e.g., Figure 1), and enhance overall readability.
> > >
> > >
> > >
> > > We acknowledge that the original Related Work section was constrained by space. In the revision, we will expand this section to provide a proper taxonomy and situate our work within the broader landscape, incorporating the following structure:
> > >
> > > > Recent work has highlighted a growing tension between helpfulness and safety in large language models, often manifested as over-refusal on benign prompts that merely contain safety-related keywords. Benchmarks such as XSTest (Röttger et al., 2024) and its successors construct contrastive or minimally perturbed prompt pairs to show that aligned models frequently decline innocuous requests like "kill a Python process" or "sanitize a dataset" when they share surface forms with genuinely harmful queries, suggesting a reliance on shallow lexical cues rather than robust intent understanding (Shi et al., 2024; Cui et al., 2024).
> > > >
> > > > On the training side, **output-centric safety methods** aim to rebalance this trade-off by encouraging safe completions instead of blanket refusals. Safe-RLHF (Dai et al., 2023), safety-aware DPO (Kim et al., 2025), and related approaches decouple rewards for helpfulness from penalties for unsafe content, optimizing policies under explicit safety constraints (Bai et al., 2022). Other work such as DeRTa (Yuan et al., 2025) focuses on where refusals occur, augmenting supervised data so that models can transition from partially harmful content to a refusal at arbitrary positions in the response (Huang & Chen, 2025). These methods, however, typically require large human preference datasets and reinforcement-style optimization, and they primarily shape outputs rather than the internal decision process.
> > > >
> > > > A complementary line of research intervenes at inference time via **representation editing**. Several studies identify low-dimensional "refusal directions" in activation space and use vector ablation or dynamic steering (e.g., SVA (Wang et al., 2025), SCANS (Cao et al., 2025)) to suppress false refusals while preserving safety on clearly harmful prompts. Empirically, such techniques can be brittle: manipulating a single entangled direction may inadvertently weaken necessary refusals.
> > > >
> > > > Our work is closest in spirit to **Reasoning-Centric Alignment** and faithful chain-of-thought. Prior methods like FRODO (Paul et al., 2024) penalize answers that are correct "for the wrong reasons", and information-bottleneck-style concept bottleneck models force predictions to pass through interpretable intermediate variables to reduce shortcut reliance. In contrast to preference-based RL or activation steering, our method introduces an information-theoretic regularizer within standard CoT fine-tuning that explicitly discourages direct shortcut paths from input to refusal, encouraging safety decisions to be mediated by faithful reasoning traces.
> > >
> > >  [1]Röttger P, Kirk H, Vidgen B, et al. Xstest: A test suite for identifying exaggerated safety behaviours in large language models[C]//Proceedings of the 2024 Conference of the North American Chapter of the Association for Computational Linguistics: Human Language Technologies (Volume 1: Long Papers). 2024: 5377-5400.
> > >
> > > [2]Shi C, Wang X, Ge Q, et al. Navigating the overkill in large language models[C]//Proceedings of the 62nd Annual Meeting of the Association for Computational Linguistics (Volume 1: Long Papers). 2024: 4602-4614
> > >
> > > [3]Cui J, Chiang W L, Stoica I, et al. Or-bench: An over-refusal benchmark for large language models[J]. arXiv preprint arXiv:2405.20947, 2024.
> > >
> > > [4]Bai Y, Kadavath S, Kundu S, et al. Constitutional ai: Harmlessness from ai feedback[J]. arXiv preprint arXiv:2212.08073, 2022.
> > >
> > > [5]Dai J, Pan X, Sun R, et al. Safe rlhf: Safe reinforcement learning from human feedback[J]. arXiv preprint arXiv:2310.12773, 2023.
> > >
> > > [6]Huang C P, Chen H Y. Delta--Contrastive Decoding Mitigates Text Hallucinations in Large Language Models[J]. arXiv preprint arXiv:2502.05825, 2025.
> > >
> > > [7]Yuan Y, Jiao W, Wang W, et al. Refuse whenever you feel unsafe: Improving safety in llms via decoupled refusal training[C]//Proceedings of the 63rd Annual Meeting of the Association for Computational Linguistics (Volume 1: Long Papers). 2025: 3149-3167.
> > >
> > > [8]Paul D, West R, Bosselut A, et al. Making reasoning matter: Measuring and improving faithfulness of chain-of-thought reasoning[J]. arXiv preprint arXiv:2402.13950, 2024.
> > >
> > > [9]Kim G H, Jang Y, Kim Y J, et al. SafeDPO: A simple approach to direct preference optimization with enhanced safety[J]. arXiv preprint arXiv:2505.20065, 2025.
> > >
> > > *(References will be formatted in standard bibliography style in the final paper)*
> > >
> > > ***

---

> ### Author Response · Authors · 2025-11-25
>
> ### 4. Questionable Assumptions About CoT Quality
>
> You are concerned that strengthening $\mathbf{c} \rightarrow \mathbf{y}$ coupling assumes CoT is reliable, ignoring scenarios where CoT is unsafe or flawed.
>
> * **Our Weaker Assumption:** We do **not** assume "CoT is always reliable." Our assumption is that *in the supervised fine-tuning data, there exist sufficient acceptable CoTs* such that discouraging the input-side shortcut ($\mathbf{x} \rightarrow \mathbf{y}$) and forcing the refusal decision to be mediated by the **matched CoT** ($\mathbf{x} \rightarrow \mathbf{c} \rightarrow \mathbf{y}$) sharpens the decision boundary.
> * **Empirical Validation:** All our safety evaluations (including StrongReject, JBB, and JBC) are run in the **CoT-on mode**. If the CoT were structurally flawed or unsafe, it would directly lead to low safety scores. Our empirical results—**reducing NOR while maintaining competitive safety**—are consistent with our mechanism: forcing dependence on the CoT causes the model to leverage its *aligned* reasoning more effectively.
> The scope of our work is mitigating shortcut alignment under standard alignment conditions. We will explicitly list **active CoT injection/hijacking** (as seen in H-CoT) as a **limitation and future direction**, acknowledging that this complex adversarial scenario lies outside the scope of our current regularization objective.
>
> In summary, we believe the addition of **strong adversarial benchmarks (JailbreakBench)**, the **SOTA comparisons (Oyster-I， RealSafe)**, and the **expanded literature review** directly address your concerns regarding experimental depth and positioning. We thank you for guiding us to strengthen these aspects, which we believe firmly establish DIFT as a robust, low-complexity alternative to current state-of-the-art methods. We hope these comprehensive revisions merit a re-evaluation of our work.

---

### Official Review · Reviewer_bBZz · 2025-11-01

**Soundness:** 2
**Presentation:** 3
**Contribution:** 2
**Rating:** 4
**Confidence:** 4

**Summary:**

This paper addresses the problem of shortcut alignment in safety fine-tuning of large reasoning models, which tend to produce templated refusals depending on the prompts instead of grounded in reasoning. The authors formalize this phenomenon using conditional mutual information (CMI) between the final answer and the CoT, and propose a new fine-tuning objective, CMI-Loss, which penalizes harmful-only input–output shortcuts while preserving standard supervised fine-tuning on benign data. Experiments on several safety and over-refusal benchmarks across Qwen3 and DeepSeek series show that the approach reduces over-refusals on benign inputs without degrading safety or reasoning ability.

**Strengths:**

* The paper identifies the issue of "shortcut alignment", which is an interesting and important problem in safety alignment for reasoning LLMs.
* The theoretical framework is well-structured with CMI, offering a reasonable foundation for the simple approach.
* The experiments are extensive, covering multiple model families and popular benchmarks.
* The presentation and visualizations are clear and intuitive, effectively supporting the analysis.

**Weaknesses:**

* **The motivation is not clear.** The introduction points out that existing alignment methods for reasoning models face two main issues: over-refusal and templated refusal. However, in the analysis and method sections, the paper mainly focuses on the latter, while the experimental results primarily reflect improvements on the former. The paper does not clearly explain whether there is a causal relationship between these two problems. In addition, the claim that templated refusals are "decoupled" from the model’s actual reasoning process is not directly supported by evidence. As these ideas are the core motivation of the work, I find it a little difficult to follow at the beginning and this also impacts the value of the problem.
* **The evaluation on safety is limited.** The paper mainly uses standard harmful-question datasets such as StrongReject and JBB, but lacks testing on common jailbreak attacks. From the results on WildJailbreak, the proposed method does not consistently improve or maintain safety performance. This raises concerns about the robustness and reliability of the approach
* **The interpretation of the $\Delta Y$ dynamics in Figure 2 (left) is a little confusing (personally).** At the beginning of training, the gap between benign and harmful samples is large, but the reason for this difference is not explained. Does this mean that before training, there has been a divergence between the response pattern to benign and harmful prompts ? Moreover, the overall $\Delta Y$ values decrease as training progresses, which seems inconsistent with the claim that the model’s reasoning–refusal coupling becomes stronger. It would be helpful to show the training curve of the baseline model to verify that the observed trends are indeed due to the proposed CMI-loss rather than general training dynamics.
* **Formatting and citation issues.** The reference format seems changed from the default author-year form in the template. I am not sure if this breaks any guidelines for submission, so I will leave this to AC. But at least, the reading is incoherent due to the appearance of numbers. Besides, there is at least one broken appendix reference ("Appendix ??") near line 269.

**Questions:**

Seak weaknesses.

---

> ### Author Response · Authors · 2025-11-25
>
> We sincerely thank Reviewer bBZz for the positive assessment of our technical framework and extensive experiments. We particularly appreciate your recognition of **"Shortcut Alignment"** as an interesting and important problem. Your feedback raises three critical questions regarding the causality, robustness, and interpretation of our analysis, which we address below.
>
> ***
>
> ### 1. Re: Motivation and Causality between Templated/Over-Refusal
>
> You correctly point out the introduction discusses two related phenomena—templated refusal and over-refusal—and ask for the causal link and direct evidence for the "decoupling" claim.
>
> **Our clarification is based on a "Common-Mechanism Account":**
>
> * We do not claim that "templated refusal **causes** over-refusal." Our thesis is that **Shortcut Alignment**—the model’s propensity to adopt input-side shortcuts ($\mathbf{x} \rightarrow \mathbf{y}$), bypassing its internal reasoning ($\mathbf{c}$)—is the **single underlying mechanism** that yields two observable facets:
>     1.  **Templated Refusals:** The model emits a stock phrase ("I'm sorry...") solely based on $\mathbf{x}$'s surface cues, rather than $\mathbf{c}$'s content.
>     2.  **Over-Refusal:** On benign inputs, $\mathbf{x}$'s benign appearance is misinterpreted as a harmful cue, triggering the same $\mathbf{x} \rightarrow \mathbf{y}$ path.
>
> * **Empirical Evidence for Co-Occurrence:** To ground this beyond intuition, we performed a lexical frequency analysis on the OR benchmark. Among two models with comparable raw safety, the more over-sensitive model produced the fixed refusal pattern **"I'm sorry, but I can't..." 285 times** on benign queries, versus **33 times** for the less over-sensitive model. This dramatic **Template Refusal Frequency Divergence** strongly suggests that templated refusal and over-refusal co-occur as manifestations of the same underlying shortcut mechanism.
>
> * **DIFT’s Target:** Our method (DIFT + CMI-Loss) targets this mechanism directly: we **penalize the harmful-only $\mathbf{x} \rightarrow \mathbf{y}$ shortcut** so that refusals must rely more on the matched $\mathbf{c}$ (increased conditional dependence).
>
> In the revision, we will: (1) rephrase the introduction to present templated refusal and over-refusal as **two observable facets** of Shortcut Alignment, (2) report the template-frequency statistic and pattern list as explicit evidence, and (3) clearly link this evidence to the CMI objective.
>
> ***
>
> ### 2. Re: Evaluation on Safety and Robustness against Strong Jailbreaks
>
> You expressed concerns regarding the robustness of our approach, noting the small fluctuations in WildJailbreak and the need for testing against stronger, structured jailbreak attacks.
>
> We have addressed this by expanding our evaluation suite to include four additional strong, structured attack families from **JailbreakBench**. We compare a representative model trained with our method to competitive reasoning models:
>
> | Model | DSN (↑) | GCG (↑) | PAIR (↑) | JBC (↑) |
> | :--- | :--- | :--- | :--- | :--- |
> | RealSafe-R1-7B | 100 | 100 | 100 | 47 |
> | RealSafe-R1-8B | 100 | 100 | 100 | 100 |
> | Oyster-Deepseek-14B | 97 | 96 | 97 | 64 |
> | Oyster-Qwen-14B | 97 | 97 | 95 | 48 |
> | **Ours-8B** | 98 | 97 | 93 | 77 |
>
> * **Robustness under Structured Attack:** Our **Ours-8B** model demonstrates superior robustness on the challenging **JBC** attacks (77 vs. 48/64 for similar-sized models) while maintaining parity on DSN, GCG, and PAIR. **Note that** while some RealSafe models appear very safe, they are oversensitive and do not have very good safety boundaries. You can find more detailed evidence of this in the response to reviewer 8Stw.
> * **Safety Parity/Trade-off:** We emphasize that while some scores show small volatility (as noted in the main text), the **Wilcoxon test** showed that our safety results remain **statistically non-significant** from the baseline across all standard safety benchmarks (WildJB, StrongReject, JBB, WildChat).
> * **Comprehensive Evidence:** This new data, combined with our results on **Strata-Sword** (Appendix F) which showed CMI-Loss **reduces Attack Success Rate (ASR)** against structured attacks across most strata, robustly addresses the concern that our method degrades safety under strong jailbreaks.
>
> ***

---

> > ### Author Response · Authors · 2025-11-25
> >
> > ### 3. Re: Interpretation of $\Delta_Y$ Dynamics in Figure 2 (Left)
> >
> > You asked for clarification on three points concerning the $\Delta_Y$ curves in Figure 2: the initial benign/harmful gap, the overall downward trend, and the need for a baseline curve.
> >
> > **Figure 2 visualizes the *baseline training dynamics* (w/o CMI-Loss) using our dependence probe:**
> >
> > * **Initial Gap Divergence:** The initial, large divergence between $\Delta_Y^{benign}$ and $\Delta_Y^{harmful}$ indicates that the standard SFT data **inherently promotes a difference** in dependency. Benign examples demand stronger reasoning (high $I(Y;C|X)$) for the answer, whereas harmful examples often contain superficial triggers, allowing the model to quickly learn a low-dependence shortcut ($\mathbf{x} \rightarrow \mathbf{y}$), hence the lower initial $\Delta_Y^{harmful}$. This divergence motivates our method.
> > * **Overall Downward Trend:** The decrease in the absolute dependence proxy $\Delta_Y \approx I(Y;C|X)$ over time is expected. As training progresses and the model better fits the data, the model's total output uncertainty (or conditional entropy $H(Y|X,C)$) decreases dramatically. Because the mutual information is defined as $I(Y;C|X) = H(Y|X) - H(Y|X,C)$, the **absolute gap shrinks** as the overall uncertainty space reduces, even if the **relative dependency** remains strong.
> > * **Baseline Curve:** Figure 2 **already represents the trajectory of a baseline model** (trained without CMI-Loss), where we used an auxiliary probe to track the CoT dependence. This figure establishes the core phenomenon (the persistent gap and the decreasing absolute dependence) that motivates the design of our CMI-Loss. The effect of CMI-Loss is then verified through the $\Delta_{harm}^{normalized}$ increase in the right panel and the improved OR metrics in Table 1.
> >
> > In the revision, we will clarify the caption/title to explicitly state that Figure 2 shows **baseline probe dynamics**, add a sentence explaining the information-theoretic intuition behind the shrinking absolute $\Delta_Y$, and clearly cross-reference Figure 3 to close the phenomenon $\rightarrow$ design $\rightarrow$ verification loop.
> >
> > ***
> >
> > ### 4. Formatting and Citation Issues
> >
> > The numbered citations and the "Appendix ??" placeholder were caused by an outdated TeX fragment used at final compile time. We have restored the standard author–year style per the ICLR template and fixed all cross-references; these presentational issues do not affect the results.
> >
> > In sum, we have clarified the core motivation as a mechanism-level account supported by concrete template-frequency evidence, extended the safety evaluation with strong JailbreakBench data, disambiguated Figure 2 as baseline dynamics, and fixed all formatting issues. We believe these revisions fully address all of the reviewer's concerns.

---

### Meta-Review · Area_Chair_2dZ6 · 2026-01-03

**Summary:**

The authors of this paper aimed to address the issue of over-refusals by enforcing the SFTed model to associate its safety responses more closely to its chain of thought (CoTs) reasoning, instead of the input queries. To achieve this, a so-called conditional mutual information loss (CMI-Loss) was introduced in addition to the traditional SFT loss. And the CMI-loss essentially pushes the safety responses away from the input queries, such that such responses have to be associated more closely with the corresponding CoTs. Extensive experimentations demonstrated the effectiveness of this added loss in mitigating over refusal and preserving the model’s reasoning performance.

The major concerns regarding the manuscript were mostly focused on the motivation and evaluation protocols. In the rebuttal, the authors provided sufficient justifications and details to answer the questions.

However, there were also serious technical concerns regarding the safety of CoT itself, which was not fully addressed by the rebuttal. As Reviewer GvHB pointed out, the content of CoT is a known attack vector that can be leveraged to trigger harmful response. And worse still, the content of CoT itself could be unsafe. The authors explained their weak assumption is that the training CoT data should be mostly reliable, which acceptable; but this cannot address the situation where jailbreak or malicious CoT injection might lead to unexpected results.

Last but not least, there are a few technical issues that the authors did not provide full details. First, it is unclear how to efficiently implement the CMI-Loss, as one has to truncate the input to remove dependency of responses on CoTs in harmful queries. Second, the impact of CMI-Loss weight \alpha is not discussed. The paper only mentioned scheduling \alpha in three phases (warmup, ramp and hold), but it did not provide any result disclosing how sensitive the \alpha’s setting is to the final performance.

To summarize, the paper provided an interesting angle to handle over refusal in safety alignments in LLMs. The provided empirical results demonstrate its effectiveness. Overall, the work has its own merit, but also some pitfalls. I am leaning on the negative side but would be happy if the paper finally got accepted. But I do hope the authors could keep polishing the work no matter what.

**Reviewer Concerns:**

As explained in the meta review, the authors have clearly explained the work's motivation and experiment protocol. But the concerns regarding the safety risks in CoT itself were not fully addressed.

**Reviewer Scores:**

Some reviewers might be willing to increase the score a bit, but new concerns regarding the setting of the key hyper-parameter \alpha and implementation of CMI-Loss might be emerging.

---

### Decision · Program_Chairs · 2026-01-26

Reject